# Digital healthcare interventions to support parents with acutely ill children at home: A systematic review

**Matthew C. Carey**[1], **Jane Peters**[1], **Anna Chick**[1], **Bernie Carter** [2]*, **Lucy Bray**[2], **Damian Roland**[3,4], **Sarah Neill**[1]

1 School of Nursing and Midwifery, University of Plymouth, Plymouth, United Kingdom, 2 Faculty of Health, Social Care and Medicine, Edge Hill University, Ormskirk, United Kingdom, 3 SAPPHIRE Group, Population Health Sciences, University of Leicester, Leicester, United Kingdom, 4 Paediatric Emergency Medicine Leicester Academic (PEMLA) Group, Children's Emergency Department, Leicester Royal Infirmary, Leicester, United Kingdom

* bernie.carter@edgehill.ac.uk

## Abstract

Short lived acute illness in children is common, yet their parents often feel uncertain about recognising signs symptoms of acute illness and knowing when to seek medical intervention. This has led to seeking unscheduled or delayed support. Digital and mobile technologies are being used to support individuals with healthcare needs, known as digital health interventions. Parents have access to digital health interventions that provide information regarding children's health, yet there is limited exploration of how these are used to support decision-making when caring for acutely ill children. This systematic review was undertaken to explore digital interventions to support parents with acutely ill children at home. Studies were identified by following the Preferred Reporting Items for Systematic reviews and Meta-Analyses (PRISMA) guidelines. A search of five databases (MEDLINE, CINAHL, Embase, PsycNET, and Web of Knowledge) was conducted using search terms (Medical Subject Headings and keywords) relating to digital interventions, children, acute illness, and health information. Forty-eight papers were screened; seven were included in the review and critically appraised using the Mixed Method Appraisal Tool. In total, 3,558 parents were included. Meta-analysis was not possible due to heterogeneity of papers; thus, narrative synthesis was used to synthesize results and explore relationships between studies. The following aspects were documented: types and characteristics of interventions; how interventions were developed; accessibility, usability and acceptability; measures of impact upon parental knowledge, confidence; and satisfaction with the intervention and usefulness. Limited evidence exists on the availability, impact and efficacy of digital interventions supporting parents caring for acutely ill children at home. Barriers exist regarding accessibility, health literacy and there is limited representation of the diverse needs of parents from different countries, cultures and

**Data availability statement:** All relevant data are within the paper and its Supporting Information files.

**Funding:** The author(s) received no specific funding for this work.

**Competing interests:** The authors have declared that no competing interests exist.

populations beyond mothers. Further research is needed to co-design and evaluate digital interventions designed with, and for, these parents.

## Author summary

Parents of acutely ill children often worry about spotting signs of serious illness in their children and knowing when to seek medical help. Digital interventions are increasingly used to help parents at these times. We wanted to find out what is already known about how useful these digital interventions are for parents. We found only seven papers worldwide published since 2014 in this area. Findings from these papers show that these interventions can be positively received by parents but also identified difficulties accessing interventions and with health literacy. Limited information was provided on the development of digital interventions in the studies included with minimal reference to involving the user; content was based on professional perspectives. With so few papers, it is not possible to draw conclusions about the efficacy of digital interventions or to address the diverse needs of different countries, cultures and parent populations. Research has not kept pace with the speed of the digital development. Our review indicates that the majority digital interventions have published no evidence of effectiveness raising questions about the impact of their introduction on parent's decision-making and use of health services.

## Introduction

Acute illness is classified as a medical condition that presents with a rapid onset, which is usually short lasting. Acute self-limiting illness in children (e.g. ear infections, urinary tract infections, bronchiolitis, and vomiting) [1] is common especially in those aged under five years [2]. The presenting features of acute self-limiting illness in children can be similar to more severe conditions, such as sepsis, which can be life threatening [3].

Children with acute illness and their parents are high users of primary, urgent, and emergency care, with frequent admissions for nonurgent presentations [2,4–6]. In particular, emergency services are more frequently used by children and young people compared to adult patients [4,5]. Attendance rates of children have shown a steady increase since 1999 [6], particularly in emergency care, with little concurrent evidence of increased severity of illness [7]. This increase in attendance is evident in other countries [8–11]. Evidence suggests that a substantial portion non-urgent attendances [12]] could have been addressed in the community [13].

Evidence suggests the prevalence of these nonurgent attendances by parents is largely underpinned by factors such as poor health literacy [8,10].

Other factors which may influence parental decisions include uncertainty and lack of confidence in recognizing signs and symptoms of acute illness [14] and/or knowing

the severity of symptoms [15], or how to navigate healthcare services [16]. Mistrust of clinicians, negative experiences, and fears of wasting clinician's time or being criticized for using services inappropriately, have also been identified as factors associated with reduced or delayed help-seeking for an acutely ill child [15–20].

The rapid growth of digital health interventions (DHIs) means there is an increasing expectation that parents will use DHIs to access digital information regarding their child's health [21]. However, the use of online information is dependent on access to digital media. Poverty, including financial constraints on access to broadband and digital literacy are key barriers towards parental access to DHIs [22,23]. Access to smartphones may also limit parental access to DHIs.

Early access to user-friendly DHIs that meet the health literacy needs of parents with an acutely ill child under five years is an urgent concern, as acute childhood illness remains a significant cause of childhood mortality in the UK [24]. Children under five in under-represented communities are more often ill and are frequent users of health services [7,19,25]. However, there is also a need to establish the nature of the evidence for their effectiveness for supporting parents caring for acutely unwell children.

However, evidence from systematic reviews is contradictory as although it shows parents do turn to the internet when seeking information regarding their children's health. the information is not always helpful. Kubb and Foran's [26] review of 33 papers spanning 13 countries found that that this can create both high levels of anxiety among parents whereas Donovan, Wilcox and colleagues's [27], review that only included three papers found that digital interventions reduce anxiety, increase confidence, and improve healthcare seeking behaviours of parents. The inclusion of only three papers in the latter work is despite the plethora of DHIs identified in Benoit and colleagues's [28] environment scan and our team's UK specific environment scan [29].

In the context of minimal research evidence concerning the impact of DHIs on parents' ability to safely care for their acutely ill children, and the ever-expanding numbers of DHIs available for parents, it is important to bring existing reviews up to date to establish the state of the evidence for their effectiveness.

Our team conducted a systematic review of interventions for parents with acutely ill children in the pre-digital era [30], marking the period prior to the first time that digital health interventions as a term was used in peer reviewed research [31]. The review reported in this paper updates the earlier review to include research reporting the impact, usability, and limitations of different types of digital educational interventions to support parents caring for acutely ill children at home – see Milne-Ives and colleagues [32] for the protocol.

This systematic review focuses on four key research questions; the first two are based on those asked in our previous systematic review [30] and the last two focus on impact and usability of DHIs.

1. How have digital interventions for parents with an acutely ill child been developed (e.g., what technologies were used, and what steps were taken in their design to ensure accessibility, usability, and acceptability)?

2. What measures are used to evaluate the impact of these digital interventions at achieving their aim?

3. How do current digital interventions impact parents' knowledge of, and experience with, managing acute illness at home and use of various healthcare services for acute childhood illness?

4. What factors influence the usability and user perceptions of these interventions?

## Methods

### Study protocol

This systematic review was structured, conducted and reported in accordance with the updated guidance from the Preferred Reporting Items for Systematic reviews and Meta-Analyses (PRISMA) [33] and the Population, Intervention, Context and Outcome (PICoO) [34] frameworks. The framework within the protocol was PICOS, however, this was changed to

 

PICoO, which proved to be more appropriate to the nature of the research reviewed. The study protocol [32] was published in JMIR research protocols, International Report Identifier (IRRID): PRR1–10.2.2196/27054.

### Study design

All types of study design published in peer-review journals in the English language were eligible for inclusion.

### Selection criteria

The PICoO framework [34] was used to select studies, this is outlined in Table 1 along with the inclusion criteria.

Studies were excluded if they did not include parents or caregivers responsible for children aged under 19 years or that targeted children, instead of parents or caregivers. We limited our search to studies that were published after 2014 for two reasons, firstly, digital technology evolves rapidly [35] and this review was concerned with the current state of the field; secondly, this review updates and expands upon a previous systematic review conducted in 2014 [30] that used two of the same research questions and similar terms, therefore studies published before 2014 were likely to have been previously captured. We excluded studies that described an intervention without evaluating it and studies not published in English as we had no capacity for translation. We did not exclude studies based on quality alone, but we have noted the quality of the research when discussing their impact. Excluding low quality studies would have reduced the comprehensiveness of the review, especially given the heterogeneity of study design.

### Search strategy

We systematically searched five electronic databases (MEDLINE, CINAHL, Embase, PsycNET, and Web of Knowledge), between January 2014 and March 2025. Key terms relating to digital interventions to support parents with acutely ill children were extracted from an initial review of the literature and used to develop the search terms and search strategy.

**Table 1. Selection of studies according to PICoO criteria.**

| Criteria | Inclusion criteria |
| --- | --- |
| Population | Parents and carers who have responsibility for children aged 0–19 years.<br>Where the child had at least one acute illness, or provided education, information, or decision support that prepare parents and caregivers for the event when a child becomes ill. |
| Intervention | Digital interventions (mobile app, web-based interventions, website or smart devices) designed to support parents with acutely ill children by improving knowledge of signs and symptoms of acute childhood illness and deterioration and decision-making about health management and/or health seeking behaviour. |
| Context | Interventions in a variety of settings, including non-clinical and clinical settings.<br>Context included where the recruitment took place and where the intervention was accessed by the parent.<br>Interventions where recruitment took place or accessed online were included. |
| Outcomes | To identify the types of digital interventions used to support parents' health literacy and care of acutely ill children and their parents. Primary outcomes were expected to include:•     Health literacy<br>•   Parental and caregiver confidence in assessing illness severity and their perceived self-efficacy in caring for their child.<br>•   Levels of anxiety regarding the child's health.<br>•   Parental and caregiver health-seeking behaviour for their child.<br>•   Levels and length of engagement with the intervention.<br>•   Parent/caregiver reported experience, including measures of acceptability, usability or satisfaction.<br>•   Other outcomes deemed relevant, for example the ability of tools to identify a seriously ill child.<br>•   Any or no comparator. |
| Study types, published from 2014 | Observational<br>Qualitative<br>Cohort<br>Randomised controlled trials<br>Studies published in English |

**Table 2. Search strategy.**

| Category | MeSH | Keywords (in title or abstract) |
|---|---|---|
| Digital interventions | Telemedicine OR Mobile Applications OR Internet-based Interventions OR Internet of Things | "mHealth" OR "mobile health" OR "eHealth" OR ((mobile OR phone OR smartphone OR cell) adj3 app*) OR web OR internet OR "online intervention" OR "web-based intervention" OR "digital intervention" OR virtual OR webpage* OR website* OR "smart device*" OR "smart medical devices" OR "smart tech*" OR tool OR resource OR program OR programme |
| Family | Child OR Infant OR Newborn OR Preschool Child OR Pediatrics OR Family OR Adolescent OR Adolescent Health OR Parents OR Caregivers OR Pregnant Women | Pediatric* OR paediatric* OR child OR children OR kid OR kids OR infant* OR newborn* OR neonate* OR bab* OR babies OR toddler* OR schoolchild OR teen* OR adolescent* OR parent* OR carer* OR caregiver* OR "foster parent" OR childminder* OR "child minder*") OR pregnan* |
| Acute illness | Acute Disease OR Childhood Disease OR Injury OR Fever OR Cough OR Whooping Cough OR Diarrhea OR Earache OR Vomiting OR Respiratory Tract Infections OR Otitis OR Croup OR Bronchiolitis OR Seizures OR Exanthema OR Mucocutaneous Lymph Node Syndrome OR Conjunctivitis OR Chickenpox OR Epiglottitis OR Tonsillitis OR Common cold OR Influenza, Human OR Pharyngitis OR Meningitis OR Status Epilepticus OR Epilepsy OR Sepsis OR Virus Diseases | (acute OR "short term" OR "short-term" adj2 (illness* OR disease* OR sickness*)) OR (minor adj2 (illness* OR disease* OR sickness*)) OR unwell OR fever* OR febril* OR cough* OR diarrh* OR rash* OR vomit* OR earache* OR bronchiolit* OR (respirator* adj2 infection*) OR otitis OR croup OR seizure* OR rash OR rashes OR exanthem* OR kawasaki* OR conjunctivit* OR "chicken pox" OR chickenpox OR epiglottit* OR tonsillit* OR influenza OR flu OR "sore throat*" OR pharyngit* OR meningit* OR epilepsy OR sepsis OR septicemia OR septicaemia OR epilept* OR headache OR "neck pain" |
| Health education | Health Education OR Health Literacy OR Help-Seeking Behavior OR Information Seeking Behavior OR Access to Information OR Decision Support Techniques OR Decision Making OR Empowerment OR Prenatal Education OR Health Knowledge, Attitudes, Practice | "Health education" OR "health information" OR "health literacy" OR "information literacy" OR "information resource*" OR "treatment seeking" OR "help seeking" OR educat* OR counsel* OR "consultation behavior*" OR "consultation behaviour*" OR (decision adj2 (aid* OR support OR guidance OR help)) OR "parent information" OR "home management" OR empowerment OR confidence OR self-efficacy OR ability OR knowledge OR understanding |

Search terms included MeSH terms and keywords relating to digital interventions, children, acute illness, and health information (Table 2). For this study, acute illness included any short-term illness, whether minor or severe. We defined digital interventions as 'any digital technologies with the aim of supporting parents or caregivers with children experiencing one or more short-term illnesses' (p4.) [32]. A confirmed diagnosis of a child's illness was not required, as the focus of this review was on how digital interventions enabled parents to respond to children with symptoms of acute short-term illness at home. Due to the complexity of the search strategy, there were a significant number of MeSH and keywords generated; a full breakdown of the search can be viewed in Table 2.

## Study selection

References were uploaded to Rayyan [36] to screen the title and abstracts, this screening was undertaken by members of research team (MC, JP, AC, SN, DR, LB, BC, PR). Studies retrieved for full text review were screened by four members of the researcher team (MC, JP, AC, SN), to determine eligibility for inclusion based on the above eligibility criteria. Any discrepancies were resolved by discussion between members of the research team, and the reasons for inclusion and exclusion were recorded (S1 Table).

When we had identified the final set of included studies, we searched their references for published papers describing the development of those interventions, and any papers published from 2014 onwards were included in the final review. The details of the screening and selection process are recorded in a PRISMA flow diagram (Fig 1) [33].

## Data extraction

A Microsoft Excel Spreadsheet was used for data extraction. Data were extracted using the following headings: (1) publication details; (2) setting/context/country of the study; (3) how were the digital interventions developed addressing

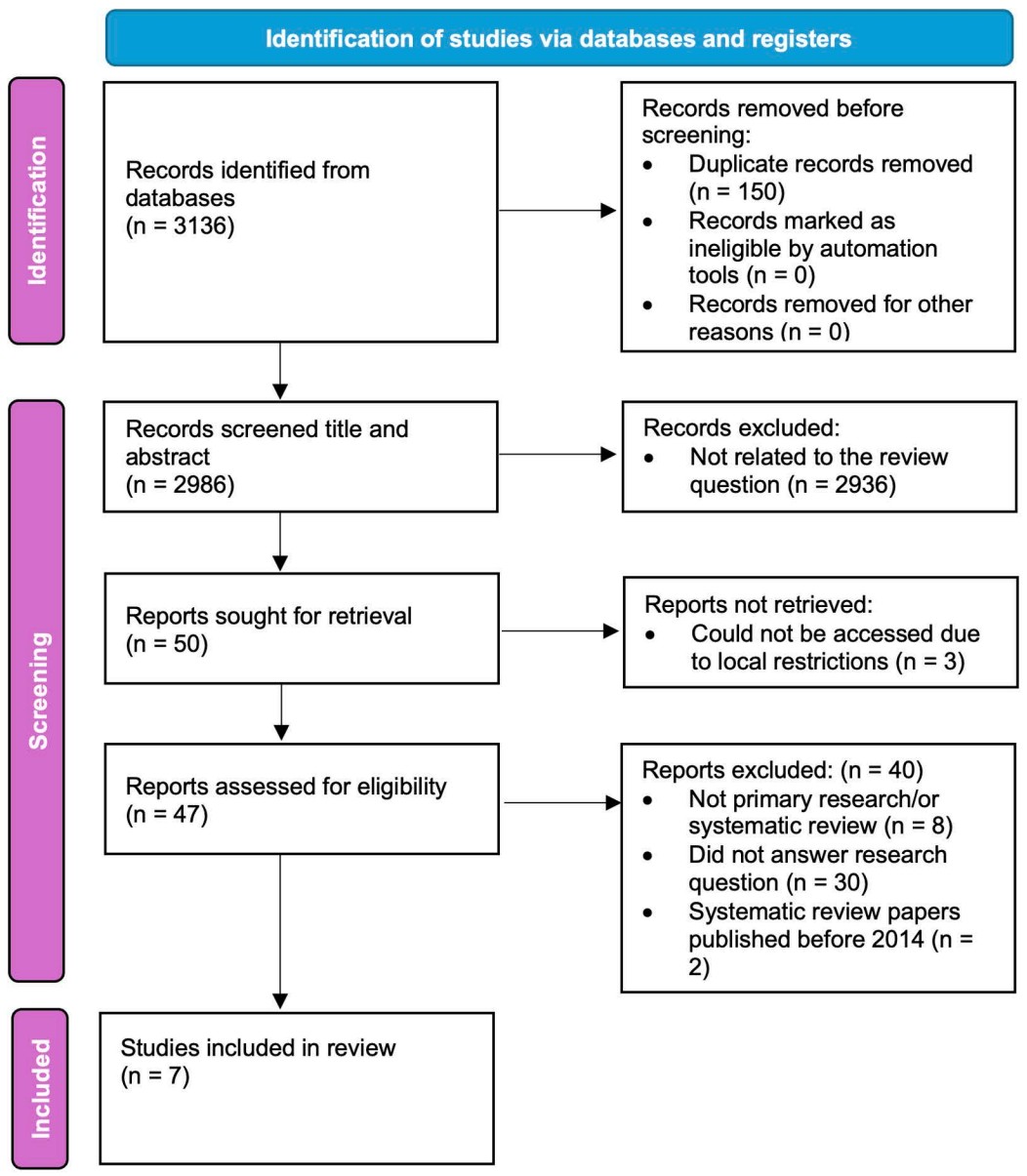

Page MJ et al. The PRISMA 2020 statement: an updated guideline for reporting systematic reviews. BMJ 2021; 372: n71. doi:10.1136/bmj.n71

**Fig 1. PRISMA flow diagram.**

accessibility/implementation, digital platform, aim and cost, intended time and place of use, age of children, type of acute illness; (4) how were these digital interventions developed addressing patient and public involvement; (5) methods used to evaluate the intervention/outcomes measured; (6) impact on parents' knowledge and experience of acute childhood illness, addressing health literacy, skills to manage child illness, parental treatment seeking behaviour; (7) impact on

parent's health service use for acute childhood illness; (8) factors identified as influencing usability, acceptability, sustainability and user perception of the intervention; (9) other key performance indicators and limitations.

Data from the included studies were extracted by one reviewer and checked by a second reviewer from the same team (AC, JP or SN, MC), any discrepancies were discussed by the review team and a consensus reached. All studies which met the inclusion criteria were included regardless of quality.

### Quality assessment

The Mixed Methods Appraisal Tool (MMAT) [37] was used for quality appraisal as the studies varied in design. The MMAT is intended to be used for systematic reviews that include qualitative, quantitative and mixed method designs [37]. Each paper is assessed according to the relevant category of their study design; 1. qualitative, 2. quantitative randomized controlled trials, 3. quantitative non-randomized, 4. quantitative descriptive and 5. mixed methods. The MMAT asks the reviewer to record a response of yes, no, can't tell, for each methodological quality criteria [37]. Not applicable (N/A) was used for categories that did not apply to the study design of the included paper. Each study was assessed for quality independently by reviewers from our team (MC, JP, AC, SN, DR, LB, BC) (Table 3).

### Data analysis

Due to heterogeneity of the research design, a meta-analysis was not possible. Narrative synthesis was used to develop a synthesis of the studies and the relationships between these studies, [38–40]. Data were analysed in a two-step process. Firstly, study characteristics were extracted according to the criteria in Table 4. In step two data was scrutinized using the criteria as outlined in the PICoO framework and made into small grouping using the PICoO template as a rubric e.g., population and characteristics [34] (Table 1). Tabulations were created where possible for descriptive and statistical data. Each grouping was reviewed and organized according to similarity and meaning. The initial sorting of the data was carried out by one reviewer (SN) and then checked independently by a second reviewer (MC); the final analysis was reviewed by all the authors. Data were presented for each component of the PICoO framework as narrative description with the support of descriptive and inferential data where available.

### Findings

Seven papers were identified which met the inclusion criteria (See Table 4): two from Ghana [41,42], two from The Netherlands [43,44] and one each from Canada [45], United States [46] and Germany [47].

A total of 3,558 parents or carers participated in these projects. Not all papers report the gender of parents or carers involved; however, it is clear that when reported the majority were mothers (n = 1,708 versus n = 304 fathers and n = 18 others) [41–43,45–47]. Most of these studies included both mothers and fathers, apart from two [41,42] that only included mothers. Some papers provided information on the age of participant's child for whom information was being sought. The age ranges of participants children varied from 0 to 18 years: 0–5 years [42,43], 0–10 years [41], 0–12 years [46] to 0–18 [44], although parents of children constituted only 25% of the latter's sample (most participants were adult patients). No information was provided on the gender of the children for whom parents were seeking information (Table 5).

Two of the papers used a mobile phone-based interactive voice response tool in Ghana [41,42]. The remaining five each used a different digital intervention, from read-only websites in Canada [45] and The Netherlands [43], a web-based module in Canada which was compared to a read-only website [45], and mobile apps in Germany - 'FeverApp' [47], The Netherlands –'Should I see a Doctor?' app [44] and the United States – 'Children's On Call' app [46]. All but two of these interventions [41,42] were designed for use without an interactive element. The Brinkel and colleagues [41] study tool, eHISS, was described as an interactive voice response (IVR) system; *a two-way conversation between a real person and a pre-recorded voice in an interactive manner.*' These pre-recorded messages had been developed and structured into an

**Table 3. Quality assessment of included studies using the Mixed Methods Appraisal Tool.**

| Methodological quality criteria | Acquah-Gyan et al. 2022 | Brinkel et al. 2017 | Hart et al. 2019 | Lepley et al. 2020 | Schwarz et al. 2021 | van de Maat et al. 2018 | Verzavoort et al. 2018 |
|---|---|---|---|---|---|---|---|
| **Screening questions (for all types)** | | | | | | | |
| S1. Are there clear research questions? | Y | Y | Y | Y | Y | Y | Y |
| S2. Do the collected data allow to address the research questions? | Y | Y | Y | Y | Y | Y | Y |
| 1.Qualitative | | | | | | | |
| 1.1. Is the qualitative approach appropriate to answer the research question? | Y | N/A | N/A | | N/A | N/A | N/A |
| 1.2. Are the qualitative data collection methods adequate to address the research question? | Y | N/A | N/A | N/A | N/A | N/A | N/A |
| 1.3. Are the findings adequately derived from the data? | Y | N/A | N/A | N/A | N/A | N/A | N/A |
| 1.4. Is the interpretation of results sufficiently substantiated by data? | Y | N/A | N/A | N/A | N/A | N/A | N/A |
| 1.5. Is there coherence between qualitative data sources, collection, analysis and interpretation? | Y | N/A | N/A | N/A | N/A | N/A | N/A |
| 2.Quantitative randomized controlled (trials) | | | | | | | |
| 2.1. Is randomization appropriately performed? | N/A | N/A | Y | Y | N/A | N/A | N/A |
| 2.2. Are the groups comparable at baseline? | N/A | N/A | Y | Y | N/A | N/A | N/A |
| 2.3. Are there complete outcome data? | N/A | N/A | Y | Y | N/A | N/A | N/A |
| 2.4. Are the outcome assessors blinded to the intervention provided? | N/A | N/A | N | N | N/A | N/A | N/A |
| 2.5. Did the participants adhere to the assigned intervention? | N/A | N/A | Y | Y | N/A | N/A | N/A |
| 3.Quantitative nonrandomized | | | | | | | |
| 3.1. Are the participants representative of the target population? | N/A | N/A | N/A | N/A | Y | N/A | N/A |
| 3.2. Are measurements appropriate regarding both the outcome and the intervention (or exposure)? | N/A | N/A | N/A | N/A | Y | N/A | N/A |
| 3.3. Are there complete outcome data? | N/A | N/A | N/A | N/A | Y | N/A | N/A |
| 3.4. Are the confounders accounted for in the design and analysis? | N/A | N/A | N/A | N/A | Partially | N/A | N/A |
| 3.5. During the study period, is the intervention administered (or exposure occurred) as intended? | N/A | N/A | N/A | N/A | Y | N/A | N/A |
| 4.Quantitative descriptive | | | | | | | |
| 4.1. Is the sampling strategy relevant to address the research question? | N/A | N/A | N/A | N/A | N/A | N/A | Y |
| 4.2. Is the sample representative of the target population? | N/A | N/A | N/A | N/A | N/A | N/A | Y |
| 4.3. Are the measurements appropriate? | N/A | N/A | N/A | N/A | N/A | N/A | Y |
| 4.4. Is the risk of nonresponse bias low? | N/A | N/A | N/A | N/A | N/A | N/A | Y |
| 4.5. Is the statistical analysis appropriate to answer the research question? | N/A | N/A | N/A | N/A | N/A | N/A | Y |
| **Mixed methods** | | | | | | | |
| 5.1. Is there an adequate rationale for using a mixed methods design to address the research question? | N/A | Y | N/A | N/A | N/A | Y | N/A |
| 5.2. Are the different components of the study effectively integrated to answer the research question? | N/A | N | N/A | N/A | N/A | Y | N/A |
| 5.3. Are the outputs of the integration of qualitative and quantitative components adequately interpreted? | N/A | Y | N/A | N/A | N/A | Y | N/A |

*(Continued)*

**Table 3.** (Continued)

| Methodological quality criteria | Acquah-Gyan et al. 2022 | Brinkel et al. 2017 | Hart et al. 2019 | Lepley et al. 2020 | Schwarz et al. 2021 | van de Maat et al. 2018 | Verzavoort et al. 2018 |
|---|---|---|---|---|---|---|---|
| 5.4. Are divergences and inconsistencies between quantitative and qualitative results adequately addressed? | N/A | Can't tell | N/A | N/A | N/A | Y | N/A |
| 5.5. Do the different components of the study adhere to the quality criteria of each tradition of the methods involved? | N/A | Y | N/A | N/A | N/A | Y | N/A |

algorithm. Acquah-Gyan and colleagues [42] evaluated a very similar tool, mHISS, which may be a later iteration of the same intervention.

Three interventions were implemented in an emergency department [43,45,46], two in non-acute hospital settings [41,43], two in primary care settings; one of which was generalist primary care [44] and the other in paediatric primary care [47]. One intervention was also delivered outside of health services in a day nursery setting [43]. It was unclear where Acquah-Gyan, Acheampong and colleagues' [42] intervention was implemented although the evaluation study was conducted in either a hospital conference room or in participants' homes.

Different methodologies were employed to evaluate the interventions, randomised controlled trials [45,46], mixed methods [41,43], descriptive qualitative [42] and prospective cross-sectional methodologies [44,47]. Data were collected in different ways, directly from the mobile app as usage data and from pop-up surveys [44,46,47], through separate questionnaires or surveys [41,43,45,46], interviews [42–44] and focus groups [41–43].

## 1.  Digital intervention development

As with studies from our previous review [30], limited information was reported on the development of the interventions. Brinkel and colleagues [41] and Acquah-Guan and colleagues [42] based their IVR pre-recorded messages and algorithm on the World Health Organisation [48] Guidelines for the Management of Common Childhood illnesses. Interventions in four projects were developed by health professionals and digital developers [44–47], three based on existing material [44,46,47] and one developed 'de novo' [45]. Only van de Maat and colleagues [43] reported involving health researchers *and* parents in the development of their web-based hospital discharge information package about fever in children with parents involved in study design, protocol development and the conduct of the study. Data from parents informed the development of their intervention. Hart and colleagues [45] included parents in pilot testing their parental knowledge questionnaire which was used to evaluate the intervention.

### Technology development: Accessibility, usability and acceptability

There was no or minimal information in the included papers on accessibility, usability and acceptability during the development of their interventions.

Accessibility was assumed in Brinkel and colleagues' [41] study in Ghana. Ghana was reported to have high levels of mobile phone usage [41], although Acquah-Gyan and colleagues [42] reported problems with unreliable mobile phone signals and electricity supply. In Hart and colleagues' [45] study 83% of households in Canada were reported to have internet access. Schwarz and colleagues [47] controlled access to their app so that it was only accessible through selected paediatric general practices for the validation of data and improvement of the app's usability. Only Schwarz and colleagues [47] reported any consideration of accessibility beyond parents' digital access to the intervention. There was no reported consideration of other aspects of accessibility such as language, literacy or learning difficulties in the *development* of the interventions [47].

Usability was only mentioned by Schwarz and colleagues [47] who reported that their app had a night mode built in and simple navigation between screens to improve usability. Lepley and colleagues [46] used a pre-existing app and no

**Table 4. Characteristics of included studies.**

| Author(s)/ Date | Setting | Aim | Design | Sample | Intervention | Main outcomes | Quality assessment |
|---|---|---|---|---|---|---|---|
| Acquah-Gyan et al. 2022 | Local hospital/ participant's home as part of the MobChild Project (Ghana) | Explore user experiences of a mobile phone based interactive voice response system | Explorative qualitative study- focus groups and in-depth interviews | 35 women who were caregivers of children under 5 from six different rural communities | Use of an interactive voice response system to seek health-care for a child on at least one occasion | Knowledge, relief of fear and anxiety – support of decision-making, positive experience of access to healthcare via IVR- lack of infra-structure a barrier | Met all the MMAT criteria for a qualitative study. Category 1 of the MMAT tool. |
| Brinkel et al. 2017 | Teaching hospital (Ghana) | Establish the barriers and factors that might determine how accepting mothers might be to using an mHealth system | Mixed methods Qualitative- focus groups Quantitative-questionnaire | 37 women all care-givers of children aged 0–10 | Use of an interactive voice response system to seek health-care for a child on at least one occasion | A useful tool but needed day to day health advice to encourage access - might be used more with an outbreak of infectious disease No data on parental knowledge | SUS (System Useability Scale) is a standardised outcome measure for useability to supplement qualitative insights. There was wide variation in SUS scores which are not explored qualitatively to understand key themes driving SUS responses and difference between those with higher and lower scores. Category 5 of the MMAT tool. |
| Hart et al. 2019 | Children's emergency department (Ontario, Canada) | Test the hypothesis that parents who access a web based interactive module would have an increased knowledge regarding the man-agement of fever. | RCT | 233 Caregivers of children aged 1 month-17 years presenting to ED with fever over 38 degrees | Group 1: web-based module Group 2: read only website Group 3: standard written of care infor-mation regarding the management of fever | Access to web-based information either interactive or read only have a significantly greater knowledge of fever management than those using the standard written information Higher user satisfac-tion with web-based information | Outcome assessors were not blinded to the intervention. Cate-gory 2 of the MMAT tool. |
| Lepley et al. 2020 | Children's emergency department, Children's Hos-pital (Wiscon-sin, US) | Determine the (1) feasibility, (2) demand, (3) accept-ability, and (4) useful-ness of mobile health (mHealth) application compared with a written intervention distributed in a pediatric emergency department | RCT | 98 caregivers with a child aged 12 and below presenting to the ED for a non-urgent complaint | Group 1: low literacy pediatric health book with video Group 2: pediatric mHealth app Group 3: both 1 and 2 Group 4: car seat safety video and handout (control) | The book was more useful than the app- there was no signifi-cant difference in the rate of ED visits in the app group, from the book or control group. | Outcome assessors were not blinded to the intervention. Cate-gory 2 of the MMAT tool. |

*(Continued)*

| Author(s)/ Date | Setting | Aim | Design | Sample | Intervention | Main outcomes | Quality assessment |
|---|---|---|---|---|---|---|---|
| Schwarz et al. 2021 | 86 paediatric and adolescent practices (n = 86) in the community (Germany) | Examine the sociodemographic characteristics of current users of the FeverApp and what feature is of interest to them according to their usage. | Prospective cross-sectional study | Installation of the app from 1,451 families: with 1,592 users during the first 14 months run of the FeverApp | FeverApp user characteristics - use of documentation and information features – a library and educational video | Watching the educational video associated with consulting the info library more intensely (26% vs. 20%) – the role of mother, having a higher level of education or being registered with the app at an earlier date was associated with the cues to consider the training video, info library and to document – those with a cell phone with android documented more | Non-interventional. The quantitative evaluation of app feature utilisation against socioeconomic characteristic provides important insights. The findings would benefit from qualitative exploration to understand the barriers to utilisation for migrants. Confounders: was based on log data and did not account for the factors associated with use or non-use which may play an important role e.g. environmental factors that influence use such as how many children were in the house, their ages, how time-poor parents were etc. What was the app not accessed - was this lack of fever or useability/ acceptability etc. but this was not the aim of the paper. Category 3 of the MMAT tool. |
| van de Maat et al. 2018 | Emergency department, non-acute hospital setting and day nursery (Rotterdam, Netherlands) | Explore parents' views on and experiences of managing their child with a fever, to assess their behaviour and needs when looking for information about fever. Develop and evaluate a hospital discharge information package about fever in children | Mixed methods Qualitative - semi structured interviews and focus group discussion Quantitative - survey | Interviews (n = 22) caregivers with children under 18 Focus group (n = 14) caregivers Survey (n = 38) caregivers | The qualitative interviews were used to inform the development of an information package to help parents identify the risk of serious illness following hospital discharge The focus group was used to assess the feasibility of the information package The survey was used to pilot the information package | Self-reported knowledge on fever increased after using the intervention – confidence in caring for their child and deciding when to see a doctor increased | Met all the MMAT criteria for a mixed methods study. Category 5 of the MMAT tool. |

*(Continued)*

**Table 4.** (Continued)

| Author(s)/ Date | Setting | Aim | Design | Sample | Intervention | Main outcomes | Quality assessment |
|---|---|---|---|---|---|---|---|
| Verzavoort et al. 2018 | Primary care (Netherlands) | To investigate whether the smartphone application "Should I see a doctor?" could guide patients in appropriate consultation at OOH clinicals by focusing on four topics: 1) app usage, 2) user satisfaction 3) whether the app provides the correct advice and 4) whether users intend to follow the advice | Prospective cross-sectional | 6,194 respondents, of which 4,456 answered to have used the app for a current medical problem, who were included in the study group 1%–25% of users were for children under 18 126 participants – group 2 were telephoned | "Should I see a doctor?" app –a self-triage tool | Intention to follow advice given by app was highest among participants told to contact GP in day-time, and associated with child under the age of 12 Experiences of the app in group 1 and group 2 were not statistically different | Difficult to extract data relating to children. Met all the MMAT criteria for a quantitative study. (Category 4 of the MMAT tool). |

**Table 5. Characteristics of participants in included studies.**

| Authors | Characteristics of parents/children: age and range in years | Parents/carers | Total participants |
|---|---|---|---|
| Acquah-Gyan et al. 2022 | Parents: Mean age 29.6 (range 15–50) | Mothers (n = 35) | 35 |
| Brinkel et al. 2017 | Parents: Mean age 30.5 (range 18–45) | Mothers (n = 37) | 37 |
| Hart et al. 2019 | Parents: Mean age 33.2 (range 18-53) with children aged 1 month to 17 years. Mean age 46.5 months. | Mothers (n = 174) Fathers (n = 42) Other (n = 17) | 195 |
| Lepley et al. 2020 | Parents: Mean age 28 (range 25–31.5) with children 12 years or younger. Median age 2 years. | Mothers (n = 86) 87.8% Fathers (n = 12) | 98 |
| Schwarz et al. 2021 | 1,592 app users in 1,452 families. Mean age 35.5 (range 14–68) | Mothers (n = 1,328), 83.4%, Father (n = 247), 15.5% | 1,592 |
| van de Maat et al. 2018 | Parents/Carer: Mean age 34 (range 19–52) with children under 5 years | Mothers (n = 48) Fathers (n = 3) Aunt (n = 1) | 52 |
| Verzavoort et al. 2018 | 6,194 app users, 25% for children 18 years or under (1,549). | 66% female users in whole sample. No information on users for children. | 1,549 |
|  |  | Total mothers (n = 1,708) Total fathers (n = 304) | Total of all participants 3,558 |

information was provided concerning how accessibility, usability or acceptability was considered during the development of the app, other than that the app had been used by other children's hospitals. Neither van de Maat and colleagues [43] nor Acquah-Gyan and colleagues [42] reported considerations of usability or acceptability during the *development* of their intervention. Verzanvoort and colleagues (2018) stated that questions in the app were evaluated by a health professional panel and the app validated by the Scientific Institute for Quality of Healthcare, Nijmegen with the clinical phrasing of questions drawn from the Dutch Triage System. They provided no information on accessibility, usability or acceptability.

The above findings do not provide sufficient detail to inform the future development of digital interventions. Table 6 demonstrates the paucity of reporting on the development of these interventions.

## 2. Measures used to evaluate the impact of digital interventions

Two studies [44,47] used app data, such as number of times downloaded, within app use of app components, and app entered data on user characteristics. Schwarz and colleagues [47] recorded use of within app components, specifically opening the video and guide information as well as documentation by the parent users. These data were then compared to user characteristics. Verzanvoort and colleagues (2018) gathered data on use of app components, user characteristics, and symptom location. Users were also asked to complete an in-app questionnaire (6,194 users) within which they were asked if they were interested in nurse telephone follow up (143 users). The in-app questionnaire asked users why they used the app, whether they intend to follow the advice provided, clarity of information, and satisfaction with the app.

Brinkel and colleagues [41] used the System Usability Scale (SUS) questionnaire and the unified theory of acceptance and use of technology (UTAUT) to structure focus groups to gather data on determinants of technology acceptance according to UTAUT and associated behaviour, use and non-use of the system, benefits and barriers to use, and factors related to system improvement. Acquah-Gyan and colleagues [42] used a similar approach stating that their interview and focus group schedule was informed by UTAUT.

The remaining studies did not use an existing scale or framework, instead opting for pre-post questionnaires, surveys and interviews. Verzanvoort and colleagues (2018) used nurse telephone calls with a subset of users (n = 143) in addition to the app data to gather simple qualitative data. During these calls users were asked about satisfaction with the app,

**Table 6. Summary of technology development reporting.**

| Authors | Accessibility | Usability | Acceptability | Stakeholder involvement | Evidence base for digital intervention content |
|---|---|---|---|---|---|
| Acquah-Gyan et al. 2022 | Reported-limited consideration | No information | No information | Users (n = 6) pilot tested | World Health Organisation (WHO) World Health Organisation (WHO) (2014) Integrated Management of Childhood Illnesses |
| Brinkel et al. 2017 | Assumed | No information | No information | No information | WHO WHO (2013) Guidelines for the Management of Common Childhood illnesses |
| Hart et al. 2019 | Assumed | Reported – limited consideration | No information | Parents (n = 12) and nurses (n = 5) pilot tested parental knowledge questionnaire | Developed 'de novo' by research team |
| Lepley et al. 2020 | No information | No information | No information | No information | Existing app used by other children's hospitals |
| Schwarz et al. 2021 | Reported-limited consideration | Reported – limited consideration | No information | No information | Clinical guidelines implied |
| van de Maat et al. 2018 | No information | No information | No information | Health researchers and parents involved throughout. | Parents' needs and preferences data gathered during the project. |
| Verzavoort et al. 2018 | No information | No information | No information | Health professionals. | Dutch Triage System |

intention to follow advice given and suggestions to improve the app. The nurse also triaged the symptoms so that comparisons could be made between app and nurse triage.

van de Maat and colleagues [43] took a mixed methods approach using qualitative data (interviews and focus groups) and quantitative surveys. Interviews gathered data to inform the development of their information package. Focus groups of parents were used in the feasibility and pilot testing of the newly developed intervention to assess parents' perceptions of useability, usefulness of the information and the format of presentation. Parents also reported the impact on their confidence in caring for their children and on seeking help from a health professional. van de Maat and colleagues' [43] survey gathered quantitative pre-post data on self-reported knowledge and confidence in caring for, and seeking help for, a child with a fever, layout, understandability and usefulness of the information package.

Hart and colleagues [45] used web-based tools to compare the impact of each intervention. Caregivers' fever knowledge acquisition was assessed using a pre-post test questionnaire modified from Chiappini and colleagues [49]. The modified questionnaire was pilot tested with five ED nurses and 12 parents of young children for content and face validity, comprehension and readability. Caregiver satisfaction was measured using a satisfaction survey modified from Kobak and colleagues [50] by removing items not related to all treatment groups.

Lepley and colleagues [46] used surveys at one, three- and six-months post-intervention to compare demand for each study arm (book alone, app and book or app alone) including self-report on whether the app was downloaded and/or book used, acceptability (ease of understanding), perceived usefulness and efficacy (parent self-report of number of ED visits). They also used 'Newest Vital Sign', a validated measure of health literacy, before delivery of interventions. Evaluation data were analysed for any association with health literacy levels.

### 3. Impact of digital interventions on parents' knowledge of, and experience with, managing acute illness at home and use of various healthcare services for acute childhood illness

The impact of these digital interventions was reported in several categories/themes: parental knowledge, parental confidence, satisfaction with the intervention and perceived usefulness, and health service use.

#### Parental knowledge

Only two papers reported the impact of the interventions on parents' knowledge of acute childhood illness [43,45] – far fewer than the nine studies in our earlier review [30]; both interventions focused on fever. Hart and colleagues [45] found

that parents with access to web-based technology in read-only or interactive module format had greater knowledge of fever management after the intervention when compared to parents who only received written and verbal information from a nurse (control group: standard care). However, no insights were presented on the nature of the standard written information provided which may have been significantly less comprehensive or alternatively more complex. Lepley and colleagues [46] did not measure the impact of the interventions trialed on parents' knowledge. In Van de Maat and colleagues' [43] study parent's self-reported knowledge of fever increased significantly (p < 0.05) after using their information package – data are not separately presented for the leaflet and the web package, so it is impossible to ascertain any differences in outcomes. Findings from Brinkel and colleagues' [41] and Acquah-Gyan and colleagues' [42] qualitative data indicate that IVR was perceived to be useful as a health education tool in child health in preparation for childhood illness, particularly for inexperienced mothers [41] or for learning about managing symptoms during acute childhood illness [42].

### Parental confidence

Only one study measured parents' confidence to care for and/or seek help for their child with a fever, both factors were reported to increase following use of their web-based resource, although no level of statistical significance is given [43]. A related finding from the qualitative data from the same study was that parents felt empowered to contact a health professional after accessing the intervention, and professionals could support their concerns by giving information about specific signs and symptoms. Although Brinkel and colleagues [41] did not ask directly about parental confidence, the mothers in their study reported that the IVR system empowered them by providing information which gave them more control over decisions about their child's health in a culture where mothers often do not have control. Similarly, Acquah-Gyan and colleagues [42] found that participants reported that the IVR assisted with decision-making concerning either health seeking for and/or care of children with symptoms of acute illness.

### Satisfaction with the intervention and perceived usefulness

In Verzanvoort and colleagues' (2018) study over 50% users rated their satisfaction with the 'Should I see a doctor' app as satisfied (46.5%) or very satisfied (9.2%) with a further 32.8% scoring satisfaction neutral. They also found that app users' satisfaction was significantly associated with users' intention to follow advice provided within the app (p < 0.001). A sub-sample of interviewees asked why they stated they were less satisfied with the app; key reasons related to not being able to enter the complete story of their child's illness, advice not being extensive enough with no link to specific information about symptoms, inconsistency of advice within the app and advice from a clinic, and that advice to contact a doctor was given too quickly. As there was no comparison group in this study, it is not possible to know if users would have been more satisfied with alternative resources.

Caregiver satisfaction with the intervention was a secondary outcome reported by Hart and colleagues [45]. The web-based module and the read only website were rated significantly higher than standard care (written and verbal information) (both p < 0.001).

Lepley and colleagues [46] and van de Maat and colleagues [43] assessed the related concept of perceived usefulness rather than satisfaction. Lepley and colleagues' [46] survey results showed no difference in usefulness between the app and/or the book on caring for a sick child provided, although the majority of parents preferred the book. The authors reported that scoring of the usefulness of the app may have been affected by '*the fact that the app was often not downloaded*'. Difficulties in downloading the app within the ED research setting were reported. They hypothesised that the usefulness of the app may have been secondary to differences in topics covered – with more covered in the app than the book. The language used within the app was reported to be more complex compared to the book. Almost half of the participants were reported to have low health literacy (measured using 'Newest Vital Sign', a validated measure of health literacy) making the complexity of language used an important factor. These were non-comparable interventions.

van de Maat and colleagues [43] used surveys to assess perceived usefulness of their information package, which parents were reported to give high scores (median score of 5/5). In Brinkel and colleagues' [41] study, mothers scored usability

as moderate, and reported that they followed advice given, they also identified barriers to using the IVR concerning ICT infrastructure, lack of familiarity with the technology particularly for older and illiterate mothers. The lack of human interaction was also considered a potential problem by 30% (n = 11) of participants, especially if a child was severely ill. Positively, Acquah-Gyan and colleagues [42] reported that the majority of IVR users liked the IVR, liked being able to choose a local language, being able to speak to a doctor and stated that they followed the advice given. However, like Brinkel and colleagues [41] poor ICT infrastructure was a significant barrier, with calls dropping out before the interaction was complete, and an unreliable electricity supply was another barrier, particularly for rural areas. Some participants reported difficulties in following the instructions to press 1 or 2 without ending the call; authors attributed this to low levels of literacy and education in the region [42].

### Health service use

Lepley and colleagues [46] was the only study to report data on health service use in the form of parent self-reported visits to ED and recorded ED use. They found no significant differences in the rate of ED visits between the app users and/or book users. Of interest is that all parents inaccurately recalled the number of ED visits within a 6-month period making recall of service use of limited value as an outcome measure. Although van de Maat and colleagues [43] found that their website increased parents' confidence in seeking help for their child, it is not clear whether the parents felt they would be more or less likely to seek help for a sick child.

Although not directly measuring health service use, Verzanvoort and colleagues (2018) asked about intention to follow the advice given in the app. This was highest when users were told to contact a GP in the daytime – a result which was associated with children under 12 years of age. This suggests that the app could increase service use for children but without data concerning intentions prior to using the app it is not possible to determine. Both studies from Ghana reported that the majority of IVR users followed the advice given to either visit a hospital or provide care at home [41,42]. Although some of Brinkel and colleagues' [41] participants also reported seeking additional help from health services or pharmacies, there were no data to show whether or not this increased or decreased health service use.

## 4. What factors influence the usability and user perceptions of these interventions?

Six factors were identified as influencing the useability and user perceptions of these interventions. These factors were the platform used, duration of app installation, presentation and format of the information, information content, personal characteristics of the user and intervention functionality.

### Platform used

Interventions were presented on a webpage, a mobile app, an interactive voice response, or in printed form (leaflet or book), the latter for comparison with the digital intervention. Findings were mixed concerning the mode of presentation of the information; from parents preferring a printed book [46] to caregivers being more satisfied with receiving information on web-based platform compared to printed information [45]. However, Lepley and colleagues' [46] app intervention contained more complex information making it non-comparable to the book. The important finding here is the need for text to be easy to understand. In Brinkel and colleagues' [41] study mothers liked having voiced information. In Schwarz and colleagues' [47] study the Android operating system was associated with high intensity users of their 'Info Library' rather than iOS.

### Duration of app installation

In one study [47] users were reported to be more likely to use the information library provided the longer they had the app installed on their phones. This may be explained by the associated cues to consider the training video and the information library or that over a longer time frame children were more likely to have a fever.

## Presentation and format of the information

The clarity of the text and the use of red, amber and green traffic lights colours were reported to be valued [43] as, interestingly, was printed information by people with lower literacy levels [46]. Parents with less experience, in particular, were reported to value videos of clinical signs and symptoms as they liked being able to see the symptoms [43] and participants who watched an educational video used the associated information library more intensely [47].

## Information content

Severity of symptoms, reliable and consistent information was positively received by participants in van de Maat and colleagues' [43] study. Schwarz and colleagues' [47] FeverApp users were reported to visit the 6/23 sections of the inbuilt 'Info Library' most often. These sections were warning signs of fever, what is fever, certificate for employers, frequency of fever measurement, accompanying symptoms and measuring fever. 'More extensive advice'(n=4) was a suggestion for improvement for Verzanvoort and colleagues' (2018) app, as was 'linking advice to background information' (n = 2). Providing these links as hyperlinks would have the effect of not overcomplicating information presented whilst still providing more detailed information for those who need it.

Mothers (32.4%, n = 12) reported that information provided by the IVR used by Brinkel and colleagues [41] was not updated regularly and could consequently be outdated. Mothers also reported wanting to be able to share the recommendations provided by the IVR with others – an option not easy to provide within a voice-based system. In Acquah-Gyan and colleagues' [42] study some participant's expectations were not met when symptoms they were interested in were not included in the IVR algorithm.

## Personal characteristics of the user

Schwarz and colleagues [47] found that those with higher level of education, aged in their 30s (62.9% n = 946), and mothers (83.4% n = 1,244) were more likely to use the FeverApp and to document more information about their child. It is unsurprising that higher level of education is associated with greater documentation, suggesting that easier to use documentation modes (e.g. voice to text recording, automated recording from wearables or AI sensors within module devices) are needed for those with lower levels of education. Use of the app's 'Info library' was highest in mothers (p < 0.0002), and for children with higher levels of fever (p < 0.001).

## Intervention functionality

Verzanvoort and colleagues' (2018) sub-set of app users (n = 126) who were interviewed were asked what improvements they would like to see in the app: an App that starts and shuts down easily, contains back buttons, ability to add photos and to be able to include more detailed information about their symptoms. They also suggested that there should be a possibility of creating multiple profiles, including personal and profiles for others. It is unclear if they are referring to parents and children. In the FeverApp parents are reported to have created one or two child profiles [47].

Brinkel and colleagues [41] used the SUS to assess usability of their IVR system. Caregiver participants, all women, rated the system as medium acceptability (SUS score of 79.3, SD + /- 7.4) indicating good useability. Mothers highlighted the importance of an easy-to-use system, available in local languages as Ghana is a multilingual country and the importance of trust in the source of information. The main barriers to using the IVR reported by both Ghanaian studies [41,42] was the state of the ICT infrastructure in Ghana to support the system and the difficulties posed to some participants who found using the technology more challenging due to lack of familiarity with it. Brinkel and colleagues [41] noted that the training provided was reported to be particularly appreciated by older and illiterate mothers [41].

## Discussion

This systematic review aimed to establish how digital interventions have been developed, identify the measures used to evaluate the extent to which the digital intervention achieved its aim, how interventions impact parents' knowledge of, and management of, acute childhood illness, impact on health service use and identify factors that influence the usability and user perceptions of these interventions. Seven papers published over 10 years (2014 and 2024) were included in the analysis, fewer than the 22 in the preceding review spanning 14 years [30]. Key observations are now presented for further discussion.

Limited information was provided on the development of digital interventions in the studies included. Only two studies report using international guidance to inform the development of the content for their digital interventions; the others did not offer clear details. Nationally and internationally the use of standard frameworks informed by evidence is encouraged when developing digital healthcare interventions (World Health [51,52]) ; the aim being to contribute towards health systems improvements. However, only four studies would likely have had access to WHO guidance.

Furthermore, the development of these technologies should involve consulting a wide range of stakeholders from the outset, including the user [52]. Our review identified only one study involving parents as stakeholders in the development of the intervention concerned [43], a feature unreported in the remaining included studies and a core recommendation of some guidance [52]. Our earlier review [30] highlights the importance of developing interventions with the end-user, noting that the effectiveness of interventions was greater where parents were involved in the development stage. Despite a plethora of literature on the development of new digital healthcare interventions there is limited research evidence of the impact of interventions designed for parents with an acutely ill child. Donovan and colleagues' [27] highlights the paucity of available evidence to support the use of digital interventions for parents/carers with acutely ill children review and they recommend involving parents as stakeholders in future research. Additionally, only global details (e.g. healthcare professionals, researchers, digital experts). This may explain the heterogeneity in the methods used to evaluate impact but may also be due to the differences in chosen methodologies within each of the included studies.

In terms of measuring impact, there was limited evidence regarding parents' preference for web-based interventions or books. It was not possible to determine whether this was related to the nature of the interventions as details on the interventions themselves were limited. In the wider literature, one randomized parallel trial explored the effectiveness of digital healthcare interventions compared to book interventions to improve health outcomes and symptoms in adults with chronic illness [53]. Interestingly, the study reported no difference between the intervention and the comparator, with both demonstrating an improvement in symptoms and health outcomes [53].

Despite limited research measuring knowledge acquisition of parents in our review, we present emerging themes on the impact of the interventions on parents' knowledge and experience in managing acute illness. Other reviews have shown either mixed results on knowledge acquisition of populations living with chronic illness using digital interventions [54] or evidence of effectiveness in increasing parents' knowledge about their children's health [55]. Despite positive reports it may not be reasonable to extrapolate findings regarding DHIs with chronic illness compared to acute illness.

Only one reviewed study [41] mentioned training for parents; none of the others reported actions to improve accessibility to support parental literacy, digital literacy, learning difficulties or reduce language barriers, despite the obvious importance of ensuring end users can access, read and understand the content. In our 2015 review, focused as it was mostly on printed materials, some consideration was given to literacy/reading age [30]. A recent scoping review of DHIs to improve access to quality primary care services, found that digital literacy status was a barrier or marker to success of the intervention [56]. Additional factors regarding digital infrastructure in rural areas was also observed in other studies where communities with limited internet access noted reduced adoption and use of digital healthcare interventions [56]. This was evident in the two included papers conducted in Ghana [41,42]. Digital infrastructure issues were also in play in a mixed-method study of young people attempting to access digital mental health interventions in rural communities in South Africa, the young people were observed to be the most disadvantaged, resulting in poorer health outcomes [57]. This

was further impeded by high cost of data, low digital literacy and additional barriers, such as restrictive religious beliefs, limited privacy, lack of translations to native language on most platforms and complicated user interfaces [57]. Mörelius and colleagues' [55] review of the effect of DHIs on health literacy of parents of children with a health condition found all five studies included used web-based portals with great success to improve parents' health knowledge and health seeking behaviour. Despite the paucity of evidence, the review identified that DHIs do have potential to improve knowledge and health literacy of parents of children with health conditions

Across the papers in our review, most participants were mothers; other evidence indicates women are the predominant population accessing digital healthcare information [58]. A retrospective study exploring the accessibility of DHIs for adults among disadvantaged populations in the USA, reported high rates of access by white, English-speaking women, and those with private insurance [59]. Similarly, in rural South Africa the use of digital health support via mobile and digital technologies to promote maternal and child healthcare was highest among mothers [60]. This aligns positively with our findings concerning parental confidence, in particular the increased empowerment of women in countries where this may be a challenge to the cultural norm [41]. In Switzerland, a cross-sectional study exploring how parents seek health information when using digital interventions, found that parents used digital media more frequently when seeking information for general health. However, when their child was acutely unwell, they sought guidance from their paediatrician, family or personal contact [58]. In our review the one study measuring parental confidence [43] found that the intervention increased their confidence in caring and seeking help for their child. Although parents could better articulate their child's illness to the doctor, this was aligned with reduced feelings of confidence in asking them for advice. This reflects [17,61] findings that the social hierarchy can act as a barrier to seeking help from health professionals.

One key point from our review is the imbalance in the target population (more mothers compared with fathers or carers); this aligns with findings from our earlier 2015 review [30]. Although DHIs have also been found to be useful tools for communication focused on improving maternal and child health for new mothers [62], there were reports of lack of familiarity with the technology, and limited health literacy prevalent in older mothers [60]. In Donelle, Comer and colleagues's [63] study investigating digital health literacy in new parents in Canada, no significant relationship was found between digital health literacy skills and usability (of which 80% were mothers). One paper in our review [47] found that those with a higher level of education and younger parents, mostly mothers, were associated with higher use of certain digital interventions. However, these types of interventions can be beneficial to vulnerable and/or rural populations, such as mothers in rural areas of India [64], highlighting the need to tailor digital interventions to meet the needs of end users. Parents in rural communities may lack access to digital resources available to new parents in more rural communities [60]. Although our review is limited to only seven papers, we identify the benefits of digital health interventions for mothers but further research is needed to establish whether these interventions are useful for fathers and other carers.

A criticism of DHIs that signifies the line between failure or success in parent satisfaction, usefulness and confidence was accessibility and the complexity of medical language. Factors that influence the usability and user perceptions of these interventions also stem from the need for text, messages and information to be easy to understand. Fitzpatrick's [65] review, focused on the use of DHIs to improve health literacy, found that the vital aspect to improving health literacy was conveying information in a language and approach that is easily understood and engaging, whilst avoiding jargon and medical terminology. In our review parents were reported to prefer clear written or audio recorded (IVR systems) information, especially for those with low literacy and less experience. Parents liked the use of interactive videos and 'traffic light' systems for presenting information within digital education interventions. These findings reflect those of our safety netting interventions research programme [66].

Traffic light systems, pictures and videos have been used in UK apps offering guidance on recognising childhood illness of parents and carers [29]. However, there is no research evidence of the impact of these interventions on parents' knowledge, confidence or service use. Traffic light systems indicating the severity of symptoms have been widely used in UK national guidance for healthcare professionals [67], although this has not been an effective tool for clinicians to detect illness of children within general practice settings [68,69].

## Recommendations for practice

The evidence reviewed is very limited and a wide range of different research methods were used, making our findings suggestive rather than conclusive. The following recommendations are tentatively proposed:

- Developers and healthcare providers need to consider the diversity of the population who will access the intervention and factors that may promote accessibility (e.g. simple language, literacy level, cultural considerations and learning difficulties) to meet the needs of users.

- Developers and healthcare providers should embrace service users (parents and caregivers) as integral to the development of DHIs so as to address accessibility, usability, acceptability, and ways for parents to evaluate the success of future DHIs.

- Interactive features (especially videos of symptoms and traffic light systems for recognising acute illness) may be help parents in their assessment of acute illness at home. However, further investigation and development of a clear evidence base for this is warranted.

- Service providers could consider providing education to support parents with low digital health literacy and lack of familiarity with technology to use the DHI.

- Service providers could consider populations in rural and/or remote areas, taking account of reliability, or otherwise, of the digital infrastructure and connectivity.

## Recommendations for future research

Further research is needed concerning the impact, usability, and limitations of different types of DHIs designed to support parents with an acutely ill child. In particular:

- Exploring the impact of co-designing interventions with parents/caregivers on effectiveness and health service use with those from diverse communities, incorporating different cultures, languages and literacy levels.

- Evaluation of the impact on parental knowledge and confidence in meeting the needs of their children with acute illness at home.

- Exploring parents' preference for the type of platform used for DHIs that support parents in managing acute illness at home taking account that these preferences may be unique to factors such as country, setting, and culture.

- Exploring the needs of family members other than mothers (e.g., fathers, grandparents and carers).

- Evaluating the impact of presenting information in different formats (e.g. written, audio, video) on parental knowledge, confidence and health service use for an acutely ill child.

- Evaluating the impact of interactive or information-only DHIs on parents' (non)adoption and continuing use of digital interventions.

    Importantly any future research should also present data concerning how the digital intervention was developed.

## Limitations

### Methodology/methods

The papers included in our review used a wide range of different methodologies and methods. Only two papers reported using a framework for assessing the usability of the digital interventions concerned. Consequently, the only possible approach for the review was an integrative narrative (qualitative) approach. Greater consistency in methodological approach should be considered for future development.

## Findings

A key research question of the review was to determine the impact of digital health interventions on parental knowledge of managing acute illness at home; however, only two papers report this outcome. The included papers also lack diversity in the countries represented and thus other factors (e.g. different cultures, religious need). Two papers were from West Africa, one from each of Canada and USA with the remaining from two non-UK European countries (despite the plethora of apps in the UK). A higher number of mothers gives some insight into this population, but this needs more detailed exploration as well as a focus on the impact of other populations responsible for caring for acutely unwell children at home.

## Conclusions

This review is built on our first systematic review [30] which reviewed research reporting interventions in written, verbal and video format – no DHIs were identified. Given the rapid evolution of digital technology, this current review was conducted to identify evidence concerning the impact of DHIs to support parents with acutely ill children published since then. Despite the plethora of apps and websites providing advice to parents in the UK [29] and internationally [28], the review revealed that there is limited evidence on the availability, impact or efficacy of digital interventions to support parents caring for acutely ill children at home. Findings from the included seven papers note successes of interventions, yet highlight barriers in the form of accessibility, health literacy and limited representation of the diverse needs of different countries, cultures and parent populations beyond mothers. Research has not kept pace with the speed of the development of apps and websites. Consequently, there is no strong evidence of the benefits, and no evidence of potential harm, of using such digital interventions to support parents with acutely ill children. These are complex interventions, consequently research is now desperately needed to co-design and evaluate digital interventions designed with, and for, parents with acutely ill children using complex intervention methodologies.

## Supporting information

**S1 Table. Exclusion table: digital interventions review.**
(DOCX)

**S2 Table. PRISMA Checklist.**
(DOCX)

## Acknowledgments

The review team would like to thank Pam Rae her help in conducting the initial searches for the review and Samantha Prime for her assistance in completing the MMAT critical appraisal of the papers included in the review. We would also like to thank the wider Acutely Sick Kids Safety Netting Interventions for Families (ASK SNIFF) team for their guidance in digital interventions as support for families with acute illness.

## Author contributions

**Conceptualization:** Damian Roland, Sarah Neill.

**Formal analysis:** Matthew C. Carey, Sarah Neill.

**Investigation:** Matthew C. Carey, Jane Peters, Anna Chick, Bernie Carter, Lucy Bray, Damian Roland.

**Methodology:** Jane Peters, Sarah Neill.

**Validation:** Matthew C. Carey, Jane Peters, Sarah Neill.

**Writing – original draft:** Matthew C. Carey, Jane Peters, Sarah Neill.

**Writing – review & editing:** Matthew C. Carey, Jane Peters, Anna Chick, Bernie Carter, Lucy Bray, Damian Roland, Sarah Neill.

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
