## [Decision Letter · Decision Letter 0]

21 May 2025

Response to Reviewers
Revised Manuscript with Track Changes
Manuscript
**Journal Requirements:**

1. Please upload separate figure files in .tif or .eps format. Also, remove the figures from your manuscript file but keep the legends.

2. We have noticed that you have uploaded Supporting Information files, but you have not included a list of legends. Please add a full list of legends for your Supporting Information files after the references list.

3. As required by our policy on Data Availability, please ensure your manuscript or supplementary information includes the following: 

4. We note that your Data Availability Statement is currently as follows: “This systematic review contains all relevant data.”.

5. Some material included in your submission may be copyrighted. According to PLOS’s copyright policy, authors who use figures or other material (e.g., graphics, clipart, maps) from another author or copyright holder must demonstrate or obtain permission to publish this material under the Creative Commons Attribution 4.0 International (CC BY 4.0) License used by PLOS journals. Please closely review the details of PLOS’s copyright requirements here: PLOS Licenses and Copyright. If you need to request permissions from a copyright holder, you may use PLOS's Copyright Content Permission form.

Potential Copyright Issues:

We do not publish any copyright or trademark symbols that usually accompany proprietary names, eg (R), (C), or TM (e.g. next to drug or reagent names). Therefore please remove all instances of trademark/copyright symbols throughout the text, including (“Microsoft Excel Spreadsheet©”) on page 14.

**Additional Editor Comments (if provided):**
**Reviewers' Comments:**

**Comments to the Author**

1. Does this manuscript meet PLOS Digital Health’s publication criteria?

Reviewer #1: Yes

Reviewer #2: Yes

Reviewer #3: Yes

2. Has the statistical analysis been performed appropriately and rigorously?

Reviewer #1: N/A

Reviewer #2: N/A

Reviewer #3: N/A

3. Have the authors made all data underlying the findings in their manuscript fully available (please refer to the Data Availability Statement at the start of the manuscript PDF file)?

Reviewer #1: Yes

Reviewer #2: Yes

Reviewer #3: Yes

4. Is the manuscript presented in an intelligible fashion and written in standard English?

Reviewer #1: Yes

Reviewer #2: Yes

Reviewer #3: Yes

Reviewer #1: In the introduction, it is not clear whether you are arguing that having children in the ER is positive or negative. I think some specificity on the appropriate times to take children to the ER would help. You should also define health literacy in your study. Line 148 mentions other factors, but only mention one specifically.

There is also a bit of confusion with the continuity from the last review. While I understand the 2014 cut off, it is not clear what about 2014 marks it as the "pre-digital era". I think if there was a bit more clarity on the state of digital interventions before 2014 (as you say they are not included in the previous review) and/or other forms of interventions since 2014, that would be helpful. In line 163, 16.63% does not seem very compelling for your argument. In line 391, the word tool in MMAT tool is redundant. I would re-word the sentence in lines 752-754, it is a bit confusing.

Reviewer #2: The manuscript “Digital interventions to support parents with acutely ill children at home: a systematic review” reviews digital interventions that have been created to help parents with acutely ill children. This is a timely and important topic, given the growing use of digital tools in healthcare decision-making, which can help patients and families. The authors previously published a review of interventions focused on the pre-digital period in 2015. In this manuscript, they are focused on 2014-2025. The authors’ aim is to provide an updated overview of the literature and to assess the effectiveness of digital interventions in supporting parents caring for unwell children.

The authors identified 48 papers, of which 7 met all inclusion criteria. Due to heterogeneity across studies, they could not conduct a traditional meta-analysis so opted to do a narrative synthesis. While I find this work important and commend the authors for their efforts, I have several concerns that limit my ability to recommend it for publication in its current form. My specific concerns are outlined below.

Abstract

1.The abstract currently lacks a clear summary of the results. Please include key details such as the number of studies assessed and included, the number of patients, and a general summary of the main findings.

2.The term “digital intervention” is not defined—please clarify this early in the abstract.

Introduction

1.Overall, the introduction provides a thorough background and justification for the study. I especially appreciate the attention to global diversity in the interventions reviewed.

2.My main concern with the introduction is that it feels overly lengthy and at times repetitive. This made it a bit hard to review. Many points are restated across different paragraphs. I recommend revising to make it more concise and remove redundant information. It would also help to break up longer paragraphs (such as the first one).

3.Consider integrating the background section into the main introduction to streamline the structure, as the current introduction essentially functions as a background section.

4.Please define digital interventions earlier in the introduction, particularly for those unfamiliar with the term.

5.It would be helpful to clearly define the research gap and the aim of the study earlier in the introduction.

Methods

1.Line 256: The wording of the inclusion criteria is difficult to follow. Also, it would be clearer if the inclusion criteria were presented in Table 1. Currently, it's unclear how the text description relates to what is shown in the table.

2.The section on study selection criteria is hard to follow as the structure is unclear and a lot of information is repeated. Making this section more streamlined would improve clarity.

3.In Figure 1, 2,936 records are shown as excluded. Please clarify the reasons for exclusion.

4.Also in Figure 1, 3 reports were not retrieved. It would be helpful to provide reasons for this.

5.Table 2 presents data quality results, but there are many “NA” entries. Please explain why certain questions were not applicable and elaborate on why some studies did not meet the data quality standards.

a.This may be addressed in the subsequent table in the Results section, but the placement is confusing. Consider consolidating or cross-referencing the two tables more clearly.

6.I appreciate that a registered protocol is cited (Milne-Ives et al., 2021), which enhances the transparency of the review process.

7.The literature search appears thorough. To increase transparency, please provide examples of the search terms used.

8.For the narrative synthesis section, please elaborate on the methods used. How were themes identified and generated?

Results

1.The authors present a thorough review of the literature, and the focus on key concepts like accessibility is appreciated.

Discussion

1.Like the introduction, the discussion is quite long and contains redundant information, which makes it difficult to follow/review. While there is valuable detail, the section would benefit from being more concise.

a.Several paragraphs are overly long and could be broken up for better readability.

General

2.In general, there are a number of grammatical errors throughout the manuscript (many in intro and methods). I recommend the authors review and edit for grammar, clarity, and language quality to improve readability.

Reviewer #3: A very thoughtful topic as per the rise in digitalization era, but there are some modifications required before the final acceptance of the manuscript. The reference list mentioned is in a haphazard manner, so kindly arrange the reference list properly with adequate spacing.

**Do you want your identity to be public for this peer review?** For information about this choice, including consent withdrawal, please see our Privacy Policy

Reviewer #1: **Yes: ** Briana M. Williams

Reviewer #2: No

Reviewer #3: No

**Figure resubmission:****Reproducibility:** To enhance the reproducibility of your results, we recommend that authors of applicable studies deposit laboratory protocols in protocols.io, where a protocol can be assigned its own identifier (DOI) such that it can be cited independently in the future. Additionally, PLOS ONE offers an option to publish peer-reviewed clinical study protocols. Read more information on sharing protocols at https://plos.org/protocols?utm_medium=editorial-email&utm_source=authorletters&utm_campaign=protocols

---

## [Decision Letter · Decision Letter 1]

14 Aug 2025

Digital healthcare interventions to support parents with acutely ill children at home: a systematic review.

PDIG-D-25-00208R1

Dear Dr. Carter,

We're pleased to inform you that your manuscript has been judged scientifically suitable for publication and will be formally accepted for publication once it meets all outstanding technical requirements.

Within one week, you'll receive an e-mail detailing the required amendments. When these have been addressed, you'll receive a formal acceptance letter and your manuscript will be scheduled for publication.

An invoice for payment will follow shortly after the formal acceptance. To ensure an efficient process, please log into Editorial Manager at https://www.editorialmanager.com/pdig/ click the 'Update My Information' link at the top of the page, and double check that your user information is up-to-date. For billing related questions, please contact billing support at https://plos.my.site.com/s/.

Kind regards,

Dukyong Yoon

Section Editor

PLOS Digital Health

Additional Editor Comments (optional):

Reviewers' comments:

Reviewer's Responses to Questions

**Comments to the Author**

Reviewer #1: All comments have been addressed

publication criteria?

Reviewer #1: Yes

3. Has the statistical analysis been performed appropriately and rigorously?

Reviewer #1: N/A

4. Have the authors made all data underlying the findings in their manuscript fully available (please refer to the Data Availability Statement at the start of the manuscript PDF file)?

Reviewer #1: Yes

5. Is the manuscript presented in an intelligible fashion and written in standard English?

PLOS Digital Health does not copyedit accepted manuscripts, so the language in submitted articles must be clear, correct, and unambiguous. Any typographical or grammatical errors should be corrected at revision, so please note any specific errors here.

Reviewer #1: Yes

Reviewer #1: The authors have addressed all of the comments that I made on the previous iteration of the article, clarifying and adding definitions in places that were specified. I do think the introduction is a bit choppy and could benefit from some more editing, but otherwise, much improved

**Do you want your identity to be public for this peer review?** For information about this choice, including consent withdrawal, please see our Privacy Policy

Reviewer #1: Yes: Briana M. Williams
